# Lateral melt variations induce shift in Io's peak tidal heating

Allard Veenstra [1] ✉, Marc Rovira-Navarro [1], Teresa Steinke[1],
Ashley Gerard Davies [2] & Wouter van der Wal [1]

The innermost Galilean moon, Io, exhibits widespread tidally-driven volcanism. Monitoring of its volcanoes has revealed that they are not homogeneously distributed across its surface: volcanic activity is higher at low latitudes and peaks east of the sub- and anti-Jovian points. Dissipation in a radially symmetric solid body cannot explain the observed longitudinal shift but dissipation in a magma ocean can. However, recent observations show that Io does not have one. Here, we demonstrate that a longitudinal shift in the heating pattern naturally arises from the feedback between tidal heating and melt production. The feedback between tidal dissipation and interior properties that results in interiors that deviate from radial symmetry is expected to drive the interior evolution of other tidally-active worlds, including icy moons such as Europa and Enceladus and exo-planets/moons with high eccentricity or obliquity.

Io is the most volcanically active body in the Solar System. The moon's volcanism is the consequence of strong tidal forces that periodically deform the moon throughout its 42.5 h long eccentric orbit (see Fig. 1)[1]. Friction within Io's interior converts part of the tidal energy into heat, which is ultimately released via volcanism. Where this heat is generated depends on the moon's interior properties. For instance, tidal heating concentrates at polar regions for a uniform mantle, while it increases in low latitudes if a low-viscosity asthenosphere is present. Because of this connection, the distribution of volcanic activity on Io's surface can be linked to the interior properties of the moon [e.g.,[2–5]]. The integration of observations spanning decades by spacecraft (Voyager, Galileo, Cassini, New Horizons, Juno) and ground-based telescopes (e.g., IRTF, Keck) has allowed the global mapping of Io's volcanic activity. Observations indicate that the activity is not evenly distributed but clusters at low latitudes (between ± 60° N) and peaks roughly 30-60 degrees to the east of the sub-Jovian and anti-Jovian points[4,6–12].

The heightened volcanic activity at low latitudes suggests that dissipation in Io occurs in a low viscosity asthenosphere[4,6,9,13] (also see Supplementary Fig. 1). However, traditional models of dissipation in a solid (or partially molten) Io cannot explain the observed longitudinal shift in volcanic activity. Alternatively, dissipation in a fully molten

magma ocean peaks east of the sub-Jovian and anti-Jovian points[14,15]. The presence of a global magma ocean in Io has long been debated [e.g.,[1,16–18]] but recent measurements of Io's tidal Love number $k_2$ refute the existence of a shallow, global magma ocean[19].

With the magma ocean hypothesis no longer considered viable, the question emerges: what drives the eastward shift? A common assumption of previous models is that Io is spherically symmetric, i.e., its interior properties are independent of longitude and latitude. However, this assumption is difficult to justify given that tidal heating itself is strongly laterally dependent. Here, we show that tidal dissipation in a non-spherically symmetric, solid Io-shaped by a naturally evolving feedback mechanism-results in an eastward-shifted pattern.

## Results

### The feedback between tidal heating and interior

Tidal heating can drive the evolution of Io's interior. The amount of tidal dissipation and its spatial distribution within Io depend on the rheological properties of rock (i.e. viscosity $\eta$ and shear modulus $\mu$) and density[2,3]. The rheological properties, in turn, depend on the temperature and melt content[20,21], which, in Io, depend on the local amount of tidal dissipation. This interdependency introduces a feedback mechanism: regions experiencing a higher amount of tidal

[1]Faculty of Aerospace Engineering, TU Delft, Delft, The Netherlands. [2]Jet Propulsion Laboratory, California Institute of Technology, Pasadena, CA, USA.
✉e-mail: a.k.veenstra@tudelft.nl

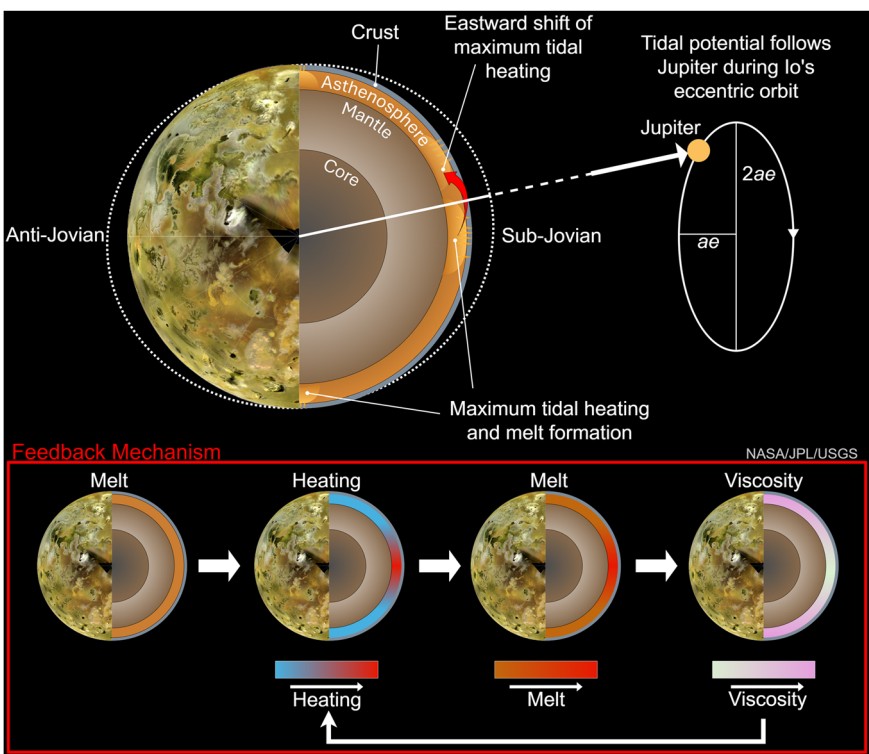

**Fig. 1 | Schematic view, looking down on Io's north pole.** Shown is the interior structure of an Io dominated by asthenospheric heating. The dotted ellipse is the equilibrium tide following Jupiter's during Io's elliptic orbit (*e* is the eccentricity of the orbit and *a* is the semi-major axis) and driving tidal heating and volcanism. The feedback between tidal heating and melt fraction (bottom) causes the tidal heating and melt pattern to evolve over time.

dissipation become warmer, decreasing the viscosity and consequently affecting the heating pattern (Fig. 1). This feedback has long been known [e.g.,[22,23]] but efficiently modeling the tidal response of a body with lateral heterogeneities has only recently become possible[24–27].

We leverage recent advances in tidal modeling and increased computational power to model this feedback for the first time. To do so, we compute tidal heating and update the interior properties recursively [see Methods]. We start the iterative process with a spherically symmetric interior model that yields an average surface heat flux of 2.24 W m$^{-2}$, consistent with observations[28,29], and iteratively solve the feedback mechanism until convergence. The feedback between interior properties and tidal heating depends on poorly understood processes, most notably heat transport in the moon's interior. To make the problem tractable, we adopt a series of simplifications.

Firstly, we only consider the feedback in Io's asthenosphere, neglecting the role of the deep mantle. This is justified as observations of the volcanic feature distribution and location and magnitude of volcanic heat flow suggest that Io is dominated by dissipation in this layer[4,6,9,12]. Analysis of 4.8 μm Juno JIRAM data showed less volcanic thermal emission (in terms of spectral radiance) from polar regions than from lower latitudes[30]. Pettine et al.[31] performed a spherical harmonic analysis on a subset of 4.8 μm JIRAM hot spot data that showed a negative correlation between their dataset and the asthenospheric heating model. However, Davies et al.[12] incorporated their 4.8 μm JIRAM dataset into an updated global database of hot spot total thermal emission. The hot spot thermal emission database was compiled from spacecraft and ground-based telescope data collected over more than three decades, including Juno data collected over seven years. The database includes hot spots that are not easily detectable by JIRAM[30]. Using this global volcanic heat flow distribution, Davies et al.[12] found a positive, albeit weak, correlation between observed volcanic

heat flow and heat flow predicted by the end-member magma ocean and asthenospheric heating (e.g., Matsuyama et al.[5]). As Io's upper mantle is believed to be close to or above the solidus temperature[32,33], we focus on the feedback between melt generation and tidal heating in Io's asthenosphere.

Secondly, we chose to simplify the complex non-linear relation between melt content (melt fraction) and tidal heating [see Methods]. We linearly couple the radially integrated tidal dissipation and melt fraction anomalies inside the asthenosphere via a proportionality constant *c* [m$^2$W$^{-1}$] that represents heat transport such as convection or melt advection[23,32,34,35]. The value of *c* depends on various physical parameters such as the average melt fraction, the permeability, and the thickness of the convective layer, and we vary *c* to account for that. Generally speaking, a larger *c* implies less blurring by lateral and radial heat transport and stronger coupling. Based on our chosen parameters, a range of possible *c* values can be inferred, representing different heat transport scenarios[23,32,35]. Within the range of allowed values, we find that we can roughly group coupling strength scenarios into four cases: weak, medium, strong, and very strong coupling strength. The precise values of *c* for these four cases can be found in the Methods.

Finally, we assume that the asthenosphere is radially uniform and has a global average melt fraction of 10% (all model parameters can be found in Table 1). While this is lower than the 20% melt fraction derived from magnetic measurements[16], this interpretation has been disputed[17,18]. Moreover, such a high melt fraction would likely lead to the formation of a magma ocean[36], which contradicts recent observations[19]. Instead, our average melt fraction of 10% is based on values found in heat transport models [e.g.,[32,37]], and we note that this still allows for local high melt fractions to conform to observations of high eruption temperatures (>1400 K)[38]. Changing the average melt fraction does not alter our conclusion as long as it is sufficiently high to support the induced melt fraction variations. The resulting melt

## Table 1 | Model parameter values

| Parameter | Notation | Value | Unit |
|---|---|---|---|
| Orbit eccentricity | $e$ | 0.0041 | - |
| Core-Mantle boundary | $R_{cmb}$ | 965 | km |
| Mantle-Asthenosphere boundary | $R_m$ | 1591.6 | km |
| Asthenosphere-Crustal boundary | $R_{ast}$ | 1791.6 | km |
| Mean radius | $R_{Io}$ | 1821.6 | km |
| Core density | $\rho_c$ | 5150 | kg m$^{-3}$ |
| Mantle, asthenosphere, and crustal density | $\rho_m$ | 3244 | kg m$^{-3}$ |
| Mantle shear modulus | $\mu_m$ | $6 \times 10^{10}$ | Pa |
| Mantle viscosity | $\eta_m$ | $10^{20}$ | Pa s |
| Asthenosphere shear modulus | $\mu_{ast}$ | $7.8 \times 10^5$ | Pa |
| Asthenosphere viscosity | $\eta_{ast}$ | $10^{11}$ | Pa s |
| Crustal shear modulus | $\mu_c$ | $6.5 \times 10^{10}$ | Pa |
| Crustal viscosity | $\eta_c$ | $10^{23}$ | Pa s |
| Average melt fraction | $\bar{\Phi}$ | 10% | - |
| Viscosity constant[a] | $B_\eta$ | 26 | - |
| Shear modulus constant[b] | $B_\mu$ | 67/15 | - |
| Proportionality constant[c] | $c$ | 0 – 0.02 | m$^2$W$^{-1}$ |

Values are taken from Segatz et al.[2] and Steinke et al.[23] unless otherwise noted.
[a]Mei et al.[20].
[b]Bierson and Nimmo[21].
[c]Value is a model parameter.

fraction patterns are converted into laterally varying rheologic parameters ($\eta$, $\mu$) using experimentally derived relations[20,21] [see Methods].

### Shift in tidal heating pattern arising from the feedback

We compute the tidal heating patterns and corresponding melt distribution for various coupling strengths, and plot the resulting surface heat flux pattern for the strong coupling strength in Fig. 2. We obtain the surface heat flux by radially integrating the tidal heating. Overlain on the surface heat flux are the locations of active volcanic features, taken from the most complete volcanic activity distribution available[12].

The feedback mechanism causes the heat flux pattern to lose its mirror symmetry around the prime meridian and have maxima that are shifted eastward. When we compare the converged heat flux pattern obtained using the strong $c$ to the heat flux pattern of the spherically symmetric case (Supplementary Fig. 1), we find that the maxima have shifted eastward by roughly 20 degrees with respect to the prime meridian (0° and 180° W, the sub-Jovian and anti-Jovian point respectively). Comparing the three coupling strength scenarios shown in Fig. 2, we see that stronger coupling (higher value of $c$) increases the longitudinal shift, increases the size of the peak-to-peak variations of the heat flux, and significantly alters the shape of the pattern. A weak coupling still produces a heat flux pattern with a longitudinal shift, but as $c$ becomes smaller, it will increasingly look like the heat flux pattern of a spherically symmetric Io.

If the coupling strength is beyond the strong coupling shown in Fig. 2, we no longer find a stable pattern, and the maxima will keep moving eastward instead. This behavior is expressed in Fig. 2 as the dotted line extending the range of possible eastward shifts of the model. The behavior might have important consequences for the dynamics of tidally heated bodies and is discussed in more detail later.

### Comparison with observations

The longitudinal shift derived from observations depends on the proxy that is used for Io's heat flux and the bin size. In subplot a of Fig. 2 we show the range of eastward shifts, as a function of bin sizes, for both the volcanic heat flux and the distribution of thermal sources[12]. The longitudinal shift for heat flux and distribution of thermal sources

differs, highlighting the challenge of comparing with observations (see Discussion).

When comparing the eastward shifts of the three models with the observed distribution of thermal sources, we find that a larger $c$ results in a better match (Fig. 2). This is evident in both the longitudinal shift and the Spearman rank correlation between predicted tidal heating and distribution of thermal sources (see Supplementary Table 1). Our strong coupling model performs the best, regardless of bin size. The match with the observed shift and the correlation generally increases with bin size.

Comparing the shift in heat flux using Fig. 2, we see a slightly larger shift on the trailing hemisphere (180°W to 360°W) but an absence of a peak in the leading hemisphere (0°W to 180°W), which our model cannot explain. The absence of a peak could be due to a dichotomy in structural properties in Io unrelated to tidal dissipation or limitations in the use of volcanic heat flux as a proxy for Io's total heat flux (see Discussion).

### Lateral heterogeneities cause the shift

Given that we started with a spherically symmetric interior, the breaking of the symmetry with respect to the prime meridian might seem surprising at first sight. However, inspection of the problem geometry reveals that such an asymmetry is not only possible but expected. The tidal potential can be decomposed into two components: the radial tide, which arises from the change in distance between Io and Jupiter; and the libration tide, which is the result of the libration of the sub-Jovian point in longitude (Fig. 1)[39]. Both components can be imagined as standing waves with a wavelength of half of Io's circumference and a period equal to Io's orbital period. While the first one is symmetric with respect to the prime meridian, the latter is symmetric with respect to the 45° meridian[39]. The response of a spherically symmetric body to these two components does not depend on where the symmetry meridian of the forcing is located. As a result, the tidal heating pattern is symmetric with respect to the 0° longitude[40]. However, this is no longer the case when lateral variations are introduced. The distribution of the lateral variations with respect to the tidal forcing pattern is different for both components (e.g., viscosity variations peaking at 0° longitude are aligned with the radial tide but not with the libration tide). This causes the response to the radial and libration tides to be distinct, which translates into an asymmetry in tidal heating.

While the emergence of an asymmetry can be expected, it is not obvious that a stable configuration should exist. To examine how the strength of the coupling between tidal heating and melt formation affects the system's evolution and the emergence of an equilibrium, we consider a toy model. For this simplified model, we consider only melt fraction variations with a wavelength of half of Io's circumference in the longitudinal direction. Such variations are represented by spherical harmonic degree 2 and order 2 (2,2), which is one of the components dominating Io's observed volcanic feature distribution[4,6].

The system's state can be represented by two input parameters: the amplitude and the eastward shift of the lateral melt fraction variations. By plotting the subsequent eastward shift of the (2,2) mode of the resulting tidal heating pattern versus both input parameters, we find that some combinations shift tidal heating patterns to the east of the input melt fraction pattern and others to the west. In between is a band of configurations without a subsequent shift, represented by the white area in Fig. 3. We can further look at the system's evolution by plotting trajectories in this parameter space, represented by the other lines in Fig. 3. Starting from a spherically symmetric model, lateral variations in melt cause the tidal heating pattern (and thus melt fraction) to migrate eastward and increase its amplitude. However, this process does not continue indefinitely. As the melt pattern shifts to the east, the increase in amplitude of the tidal heating pattern becomes

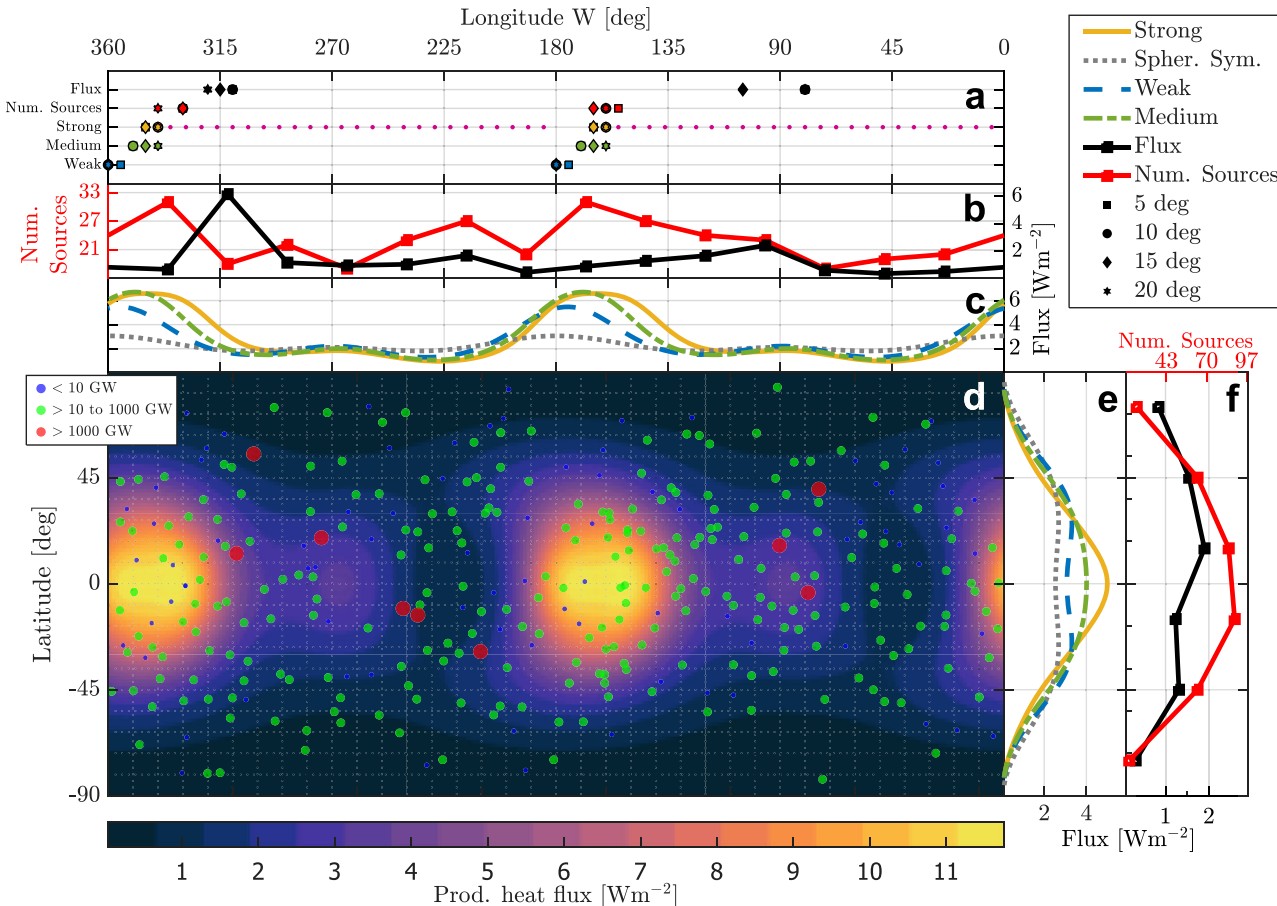

**Fig. 2 | Surface heat flux after convergence of the iterative procedure for a *c* that describes a strong coupling strength (subplot d).** Overlain are the 343 thermal sources identified by Davies et al.[12] with the marker size indicating their thermal output. Directly flanking the map are the longitudinally and latitudinally integrated fluxes from the converged solution (subplots c and e), a spherically symmetric Io, and the fluxes for two other values of *c*: a weak coupling and a medium coupling. The longitudinal and latitudinal profiles of the observed volcanic heat flux and the distribution of thermal sources[12] are shown in the subplots b and f with red and black solid lines. Subplot a shows the eastward shift for binned versions of the longitudinal profiles for different bin sizes. The bins are spaced such that the westernmost bin has its center at the prime meridian. The magnitude of the shift is taken as the longitudinal location of the highest bin on each hemisphere. The melt fraction pattern causing the plotted surface heat flux pattern is shown in Supplementary Fig. 2.

smaller, slowing down the eastward shift until it reaches a stable configuration. The decreasing amplitude of the tidal heating pattern with eastward shift can be seen in Supplementary Fig. 3. At low enough *c* values, the system eventually migrates towards the stable configuration. Because the amplitude of the melt fraction variations is tied to the value of *c*, the termination point of the iterations moves to larger eastward shifts in Fig. 3 as the value of *c* increases, as observed in Fig. 2. We find that the final, stable pattern is only determined by the value of *c*, irrespective of the starting configuration.

**A continuously moving dissipation pattern**

If we keep increasing the value of *c*, we find that the trajectory in the parameter space of Fig. 3 can move to such large eastwards shifts that it goes beyond the stable band. In that case, because of the coupling strength, the amplitude of the melt fraction variation grows so much that the decrease due to the eastward shift cannot overcome it. Thus, the peak tidal heating will keep moving eastward, periodically switching between two patterns. The two patterns are shown in Supplementary Fig. 4, and the full trajectory in parameter space is shown in Supplementary Fig. 6. Supplementary Figs. 4 and 6 also demonstrate that the system returns to its starting point in parameter space. So, while the pattern keeps changing and moving eastwards, the pattern repeats itself.

For the full model, an increase in *c* also results in continuous eastward movement, but with different spatial patterns, shown in Supplementary Fig. 5. If the conditions for a continuously moving tidal dissipation pattern occur on Io, it suggests that the current volcanic heat flux distribution is relatively recent and will not remain fixed. The continuously moving pattern could, in principle, explain any observed eastward shift in the currently observed heat flux pattern.

The timescales in our model are not well established, causing the speed at which the patterns revolve around Io to be difficult to estimate. However, from dimensional analysis, the timescale corresponding to one iteration can be roughly estimated as ~ $10^5$ yr (see Methods). Given that it takes tens of iterations for the pattern to move around Io, the timescale of the revolution of the pattern around Io is on the order of millions of years. Coincidentally, this is close to the estimated surface age of Io, at ~ 1 million years[41–43]. It might therefore be possible that past peaks in tidal heating have left geological footprints that are still visible. A potential way to test whether Io exhibits this behavior is discussed in the Discussion.

One last thing to note is that the average tidal dissipation during the periodic behavior of the toy model only varies by roughly 11%. So even though the heating pattern is dynamic, the average heating would be relatively stable and is still compatible with the recent finding that volcanism on Io has likely been relatively constant since Io's formation[44].

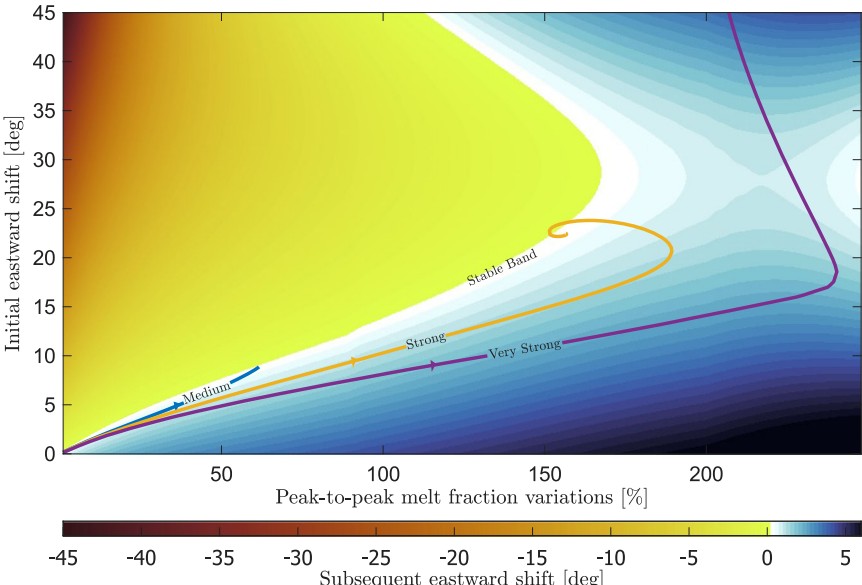

**Fig. 3 | Initial and subsequent shift versus melt fraction amplitude.** The color-map shows the subsequent longitudinal shift of the tidal dissipation pattern resulting from an imposed degree and order 2 melt fraction pattern with varying melt fraction amplitude (x-axis) and eastward shift (y-axis). We define the subsequent shift as the total shift of the tidal dissipation pattern (only the degree and order 2 part) minus the shift of the imposed melt fraction pattern. In the stable band, the white area, further iterations do not change the shift. The colored lines represent the path followed in successive iterations when we only propagate the degree and order 2 component for different coupling strengths; the corresponding values for $c$ can be found in the Methods. The case for which iterations do not result in a converged pattern (rightmost purple line) but a continuously moving pattern, is plotted in full in Supplementary Fig. 6.

## Discussion

We demonstrate that the feedback between melt generation and tidal heating drives the interior evolution of Io. Incorporating this feedback mechanism results in heat flux patterns with a peak heating that is shifted eastward, in agreement with the observed eastward shift in volcanic heat flux and distribution of thermal sources. This shift was previously explained by the existence of a magma ocean[14], but recent gravity observations have ruled this out[19]. Limitations in both our model and the observations preclude us from obtaining a complete picture of Io's interior evolution and volcanism.

Our assumption that Io only has tidal heating in the asthenosphere causes us to predict a bigger contrast between heat flux at high and low latitudes than what is seen in the observations. In reality, it is likely that a fraction of tidal heating also occurs in the deep mantle, which peaks in polar regions[2,3]. The role of additional deep mantle heating can be explored using our model, but to do this properly would require a more complicated thermal model. Nevertheless, a different starting interior model would not alter our conclusion. A discussion about the parameters of our initial interior model can be found in the Methods section.

A second simplification is the linear relation between the tidal dissipation and the melt fraction anomaly. Our model qualitatively illustrates how the tidal heating-melt fraction feedback can drive the evolution of Io. Making more quantitative predictions requires a more sophisticated method of coupling tidal heating to melt generation and temperature. Related to this are simplifications in modeling the radial and lateral transport of heat and melt. We have radially integrated the tidal dissipation pattern to create a surface heat flux but, in reality, the surface heat flux will be affected by unknown crustal and chemical heterogeneities and emplacement inside the crust[34,37]. We approximate the lateral transport of melt by removing short-wavelength features, following Tackley[45] and Tackley et al.[22]. Future studies aimed at matching the observed volcanic distribution should consider melt-migration[34,37] or convection[22] in a more realistic way. While there are many avenues for model improvement, the conclusion of an eastward shift due to the introduction of lateral variations is unlikely to be affected by our model choices.

The comparison between predicted tidal heating patterns and observations presents important challenges. We use, as have others [e.g.,[6,8,9,12]], both volcanic heat flux and distribution of thermal sources as a proxy for global heat flux. Both approaches have strengths and weaknesses. Volcanic heat flux is the most direct observation of Io's heat flux distribution, but it at best represents a snapshot of tidal heating over the last ~ 40 years, starting from 1979. This is a very short time compared to typical volcanic timescales on Earth. The distribution of sources records volcanic activity on much longer timescales (~ 1 million years[41]) but is not necessarily tightly linked to current volcanic activity, and may not be sensitive to global change on a shorter timescale. Additionally, the distribution of sites of ongoing volcanic activity skews the results towards areas with numerous small sources over features like Loki Patera, a single source that outputs orders of magnitude more energy than the sum of many smaller sources that may be clustered.

Complicating things further is the fact that infrared measurements of volcanic thermal emission total $58 \pm 1$ TW[12], which is about half of Io's global thermal emission of $105 \pm 12$ TW from infrared observations[28] and $93 \pm 19$ TW from astrometric observations[29]. The total heat output (the endogenic heat flow) is what our model and all other tidal response models[2,14,23,46] are calibrated to and display. This problem of unaccounted-for heat flux distribution could explain the mismatch between the observations and the predicted heat flux in the leading hemisphere (0° W to 180° W), especially between 135° W and 180° W, where the model predicts much more heat flux. The heat flux that is currently unaccounted for may be emanating from this area. It may be that there is a higher proportion of intrusive[34] rather than extrusive volcanism taking place here. Also not well understood is the transport of heat via gas-phase stealth volcanism[47], which is, by its nature, extremely difficult to detect in this and other regions[48]. The hypothesis of stealth or intrusive volcanism is somewhat supported by the large number of thermal sources around 150° W (see Fig. 2). Thus,

knowledge of the distribution of the other half of Io's heat output would be very beneficial in further linking our model outcomes and other tidal response models to the observations. Recent measurements by Juno's Microwave Radiometer instrument[49] and ALMA observations similar to what has been accomplished at Ganymede[50] might shed light on Io's endogenic heat flow. An alternative explanation for the lack of volcanic heat flux on the leading hemisphere could also be a leading-trailing hemisphere dichotomy in interior properties affecting tidal heating, such as crustal thickness variations[10,12]. With the model we have developed, the impact of different kinds of lateral variations can be investigated.

The introduction of lateral variations also affects the tidal response of a body[24,26,27,51]. The lateral variations found in the strong coupling model in Fig. 2 cause a difference between $k_{2,2}$ and $k_{2,0}$ on the order of 1%, which could be detectable by a future mission. For instance, this difference is above the predicted sensitivity of the Jupiter Icy Moons Explorer (JUICE) spacecraft[52], which will orbit Ganymede. However, it is on the edge of the predicted sensitivity of Europa Clipper[53] and below that of the proposed Io Volcano Observer (IVO)[54].

Finally, our model also presents a tentative explanation for any observed eastward shift, namely that the conditions in Io are such that no stable tidal dissipation pattern arises. If Io's tidal dissipation pattern is indeed continuously moving eastward, one would expect active volcanic features to be generally located eastward of old features. Hamilton et al.[6] compared the distribution of cold patera floors (volcanic features with no observed thermal output) to the distribution of active volcanic features. Their Fig. 3 suggests that active features are located more eastwards. A similar conclusion can be drawn from Steinke et al.[4] by comparing their Fig. 1 and Fig. A1.

While we have focused on Io, the feedback mechanism described above might be ubiquitous in tidally active bodies. It has long been hypothesized that variations in tidal heating can induce shell thickness variations in icy moons with subsurface oceans[55]. Lateral heterogenities will affect the tidal response, thus affecting tidal heating patterns and shell geometry. This might be especially relevant for Enceladus, whose ice shell thickness varies from ~ 30 km at the equator to a few kilometers at the South Pole and exhibits a well-known north-south dichotomy in geological activity and shell thickness[56]. A recent paper on Enceladus highlighted the importance of incorporating lateral variations[57]. Kang and Flierl[57] demonstrated that an initially small polar dichotomy might be amplified by the feedback between shell thickness and tidal response. The feedback mechanism investigated here is also expected to drive the evolution of tidally heated exoplanets and exomoons. Even if the planet or moon starts out as radially symmetric, the tidal dissipation will create hotter regions, offset from the sub-planet point, that could promote volcanism and outgassing, which could be observed. The feedback might be amplified in close-in exoplanets, which experience extreme near-side far-side temperature gradients that have been hypothesized to drive hemispherical convection cells and which promote habitability[58]. With the observations of tidally heated exomoons and exoplanets within technological reach[59–61], Io remains the best laboratory to test theories aimed at understanding the dynamics of this class of planetary bodies.

## Methods
### The tidal response of Io
We compute the tidal response of Io using *LOV3D*[27,51,62]. It solves the mass, momentum, and Poisson's equation for a self-gravitating viscoelastic body in the spectral domain using tensor spherical harmonics. Due to the inclusion of lateral variations, tides excite an infinite set of spherical harmonic modes, yet their amplitude quickly decreases with the order of the coupling (see Rovira-Navarro et al.[27] for details). We only include first-order modes, as our tests have shown that only using the first-order modes is nearly identical to also including second-order modes. The elastic component of *LOV3D* has

been benchmarked against a FEM model[63]. For the viscoelastic component, we verified that the total tidal heating computed by using the strain rate and stress tensor and the work done by the tidal force are equal (see[27]) and compared the tidal heating pattern with those produced by a FEM code[35] finding good agreement (see Supplemenatry Disscusion and Supplementary Fig. 9).

### Initial interior model
We assume that Io is incompressible, has a Maxwell rheology, and an interior model based on Segatz et al.[2,23] that produces the observed thermal output (Table 1). We chose Maxwell rheology for its simplicity and to conform to the previous model[35], but it does not match the recently obtained real part of $k_2$. The use of a more realistic rheological model, such as Andrade rheology, would allow us to match both the real and imaginary components of the $k_2$ Love number[19] without altering our conclusions, as the effect of melt fraction on Andrade rheology is the same as for Maxwell rheology[21].

It is possible to build an interior model leading to the asthenosphere heated end-member pattern using different values for the viscosity, shear modulus, and thickness of the asthenosphere[2,3,14]. The combination of parameters we chose corresponds to an asthenospheric heating pattern, as introduced by Segatz et al.[2]. Changing the starting interior model would not alter our conclusion of an eastward-shifted peak but could alter our resulting surface heat flux pattern.

As noted in the main text, we consider all tidal heating to occur in an asthenosphere. Tidal heating in the deep mantle can affect tidal heating patterns; for instance, by promoting tidal heating at the poles. While the feedback between tidal heating and mechanical properties in the deep mantle can also drive the mantle evolution and produce an eastward shift[27], we focus on the asthenosphere.

### Tidal heating-rheology feedback
The underlying idea behind our method is straightforward: an increase in dissipation should lead to an increased melt fraction. However, the relation between the tidal dissipation distribution $Q(\theta, \phi)$, in units of Wm$^{-2}$, and the melt fraction distribution $\Phi(\theta, \phi)$ is complicated. It is non-linear[21,22,37] and depends on numerous parameters such as the type of heat transport (magmatic and/or convective), the latent heat of the rock, the viscosity and thickness of the convective layer, the effectiveness of melt transport[23,32–34], and the solidus and liquidus temperature[64,65].

Instead we simplify the relation between $Q(\theta, \phi)$ and $\Phi(\theta, \phi)$. We linearly couple the anomaly of the dissipation, $\delta Q(\theta, \phi)$, to the anomaly in the melt fraction distribution, $\delta \Phi(\theta, \phi)$, as

$$\delta \Phi(\theta, \phi) = c \delta Q(\theta, \phi), \tag{1}$$

where $c$ is the proportionality constant, in units of m$^2$W$^{-1}$[35]. The anomaly in the dissipation defined as $\delta Q(\theta, \phi) = Q(\theta, \phi) - Q_{ref}$ and the anomaly in the melt fraction distribution defined as $\delta \Phi(\theta, \phi) = \Phi(\theta, \phi) - \Phi_{ref}$. Here $Q_{ref}$ and $\Phi_{ref}$ are the spatially-averaged tidal heating and melt fraction, respectively. The relation between tidal dissipation and melt fraction will only be used in the asthenosphere, conformable to our assumption that all the dissipation happens in the asthenosphere. The physical interpretation of our underlying idea and thus Eq. (1) is that we assume that the entire asthenosphere is just above the solidus temperature. Thus, any changes in the dissipation immediately lead to changes in the local melt fraction.

The melt fraction distribution is converted into a viscosity and shear modulus distribution using experimentally derived relations. Because of our assumption of an asthenosphere just above the solidus temperature, we make viscosity and shear modulus dependent only on melt and not temperature. For the viscosity, the relation is[20]

$$\eta(\theta, \phi) = \eta_{asth, \Phi} \exp\left(-B_\eta \delta \Phi(\theta, \phi)\right), \tag{2}$$

with $\eta_{asth,\bar{\Phi}}$ the viscosity of the asthenosphere for a given average melt fraction and $B_\eta = 26$ a constant which is experimentally derived[20]. This relation and the value for $B_\eta$ are likely valid up to a melt fraction of roughly 30%[66], which is above our assumed melt content of 10%. The shear modulus is computed using[21]

$$\mu(\theta,\phi) = \mu_{asth,\bar{\Phi}} \frac{1 + B_\mu \bar{\Phi}}{1 + B_\mu(\delta\Phi(\theta,\phi) + \bar{\Phi})}, \quad (3)$$

with $\mu_{asth,\bar{\Phi}}$ the shear modulus of the asthenosphere for a given average melt fraction and $B_\mu = 67/15$ from Bierson and Nimmo[21]. The values for both $B_\eta$ and $B_\mu$ are not strongly constrained, which we account for by using a range of values for $c$. The obtained viscosity and shear modulus maps can then be used to recompute the tidal response, now with the introduction of lateral variations. A schematic of our steps in the feedback mechanism can also be found in Fig. 1.

We determine the value of $c$ by assuming two end-members of heat transport models: convection and melt advection. Using convection scaling models[67], Steinke et al.[23] showed that $c$ can be approximated as

$$c \approx \frac{\Delta T_{asth}}{(T_{liq} - T_{sol})\Delta Q}, \quad (4)$$

where $\Delta T_{asth}$ is the peak-to-peak variation in temperature in the asthenosphere as found in Steinke et al.[23], $T_{liq}$ is the liquidus temperature, $T_{sol}$ is the solidus temperature, and $\Delta Q$ is the peak-to-peak variation in tidal heating.

Alternatively, $c$ can be estimated by balancing melt production and melt migration (see equations 1–5 as found in Moore[32]), from which it follows that melt fraction and tidal heating are related as $\Phi^n = Cq$. Here, $C$ is a constant containing various parameters, $q$ is the volumetric heat production (in Wm⁻³), and $n$ is 2 or 3 and comes from the permeability, which goes as $\propto \Phi^n$[32]. Differentiating both sides around an equilibrium melt fraction, $\Phi_0$, and rearranging gives $\delta\Phi = \frac{\Phi_0}{nq_0}\delta q$. Turning the volumetric heat production $q$ into surface heat flux $Q$ we get a similar relation as Eq. (1), with

$$c = \frac{\Phi_0}{nQ_0}. \quad (5)$$

Both approaches render similar $c$ values. Using typical values given in Steinke et al.[23] to fill in Eq. (4), we get that $c$ is in the range of $10^{-2}$ m²W⁻¹, with an upper limit of 0.06. Equation (5) gives similar values for $c$. Given the uncertainty in $c$, we treat it as a model parameter. To create Fig. 2 we used $c = 0.00925$, $c = 0.00975$, and $c = 0.0105$ for the weak, medium, and strong lines respectively. A $c$ larger than roughly 0.011 will result in solutions that move eastward continuously. In Fig. 3 we used $c = 0.0125$, $c = 0.015$, $c = 0.02$ for the medium, strong, and very strong lines respectively.

Heat transport is expected to blur tidal heating patterns[22,23,68]. We take this into account by omitting higher-degree spherical harmonic components of tidal dissipation when we compute the melt fraction pattern. We limit $Q(\theta,\phi)$ in Eq. (1) to only contain modes up to degree $N_{max}$ such that $Q(\theta,\phi)$ is defined as

$$Q(\theta,\phi) = \sum_{n=0}^{N_{max}} \sum_{m=-n}^{n} \dot{e}_n^m Y_n^m(\theta,\phi), \quad (6)$$

with $\dot{e}_n^m$ the tidal dissipation per degree $n$ and order $m$ as defined in Eq. 32 of Rovira-Navarro et al.[27] and $Y_n^m(\theta,\phi)$ the normalized spherical harmonics of degree $n$ and order $m$. The cut-off degree $N_{max}$ implicitly gives $c$ a lateral dependency. Limiting the number of propagated energy modes is also a practical concern. The time it takes to compute the tidal response of a body with lateral variations

scales with the amplitude and spatial scale of the lateral variations. We have varied $N_{max}$, as shown in Fig. 3, but that does not change the conclusion of an induced eastward shift. Increasing $N_{max}$ has a similar effect during iteration as increasing $c$ because the higher order modes will increase the peak-to-peak lateral variations. To generate Fig. 2, we have set $N_{max}$ to 4, consistent with the tidal dissipation pattern of a spherically symmetric Io, which contains only terms up to degree 4[40]. The surface heat flux that is plotted in Fig. 2 is the total surface heat flux pattern containing all the $\dot{e}_n^m$ modes that *LOV3D* computes. We plot the surface heat flux computed using all modes to avoid implying ways of heat transport through the crust.

### Iteration of the feedback mechanism
We model the feedback mechanism through iteration, repeating the steps in the previous section (see also Fig. 1). The introduction of lateral variations in rheology causes tidal heating to change from iteration to iteration, exceeding the average dissipation observed. This can be corrected by changing the starting, spherically symmetric interior so that its average dissipation is lower or, alternatively, by normalizing the resulting heat flux. We do not correct for this during iterations so as not to stifle a potential thermal runaway. Normalizing the dissipation to a fixed average during iteration leads to a similar behavior as not normalizing it, but the value for $c$ at which a certain eastward shift is found generally increases (see Supplementary Fig. 7).

While the iteration in our model qualitatively illustrates the system evolution; it should not be taken as the temporal evolution, as it takes time to heat the material and produce melt. We can roughly estimate the time interval corresponding to each iteration as the time it takes to melt 10% of the rock, a conservative scale for the variation in melt fraction from Supplementary Fig. 2. Using the latent heat, $L = 5 \times 10^5$ J kg⁻¹[32] and a volumetric heating of $q = 5 \times 10^{-5}$ W m⁻³ we estimate time as $t \sim \frac{\rho L}{q} \approx 1 \times 10^5$ years. A similar number can be found from dimensional analysis of $c$, using $Q_{ref}$, $L$, $\rho_m$, and $H = 200$ km, the thickness of the asthenosphere, as relevant properties.

To prevent the system from overshooting, we average the resulting melt fraction distribution with the melt fraction distribution from the previous iteration. This approach results in a continuous evolution of Io's thermal state but still allows for a quick evolution of the tidal dissipation pattern, thus saving computation time. In this way, we also prevent any unrealistic, sudden changes that can otherwise lead to non-physical oscillations around an equilibrium solution or force Io's interior to evolve into a different equilibrium, which would not be reached in realistic circumstances.

The simulation of the feedback can be summarized as follows (see flowchart in Supplementary Fig. 8): We start by computing the tidal response of a spherically symmetric Io. From the computed tidal dissipation spectrum we take the contributions up to and including degree $N_{max}$ to go into the tidal dissipation distribution. The anomaly of the resulting dissipation distribution is computed by subtracting the average and multiplying that by $c$ to yield the melt fraction anomaly. We average the melt fraction anomaly with the previous iterations' melt fraction anomaly to mimic thermal inertia. The averaged melt fraction anomaly is then used to compute the viscosity and shear modulus. Finally, we use the rheology distributions to recompute the tidal response and iterate until convergence. Convergence is defined as no more changes in the average tidal heating. If the model is in the regime of continuous eastward motion, we define convergence as the moment the average tidal heating repeats itself.

### Data availability
The compilation of observed volcanic sources used in this study is available in the Appendix of Davies et al.[12].

## Code availability

The code used to generate the tidal response (*LOV3D*) is available Rovira-Navarro[62].

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

## Acknowledgements

We thank Francis Nimmo and an anonymous reviewer for their insightful comments and suggestions during the preparation of this article. This publication is part of the project "Tidal dissipation heating of Io" with project number ENW.GO.001.042 of the research programme "User Support Programme Space Research" which is (partly) financed by the Dutch Research Council (NWO), awarded to W.W.. M.R.N. was partially supported by the National Aeronautics and Space Administration (NASA) under Grant 80NSSC23K1276. Part of this work was performed at the Jet Propulsion Laboratory, California Institute of Technology, under Government contract. A.G.D. is supported by award 80NM0018F0612 from the NASA New Frontiers Data Analysis Program.

## Author contributions

A.V., M.R.N., T.S. and W.W. conceived the idea and designed the numerical experiments. A.V., M.R.N. and W.W. analyzed the data. M.R.N. and A.V. developed the numerical model. A.V. and T.S. performed the numerical experiments using the spectral and FEM code, respectively. A.G.D. advised on the link to the observations. A.V., M.R.N., A.G.D. and W.W. wrote the manuscript. All authors contributed to the discussion of the work.

## Competing interests

The authors declare no competing interests
