## [Transparent Peer Review file · Nature Communications]

Lateral Melt Variations induce Shift in Io's Peak Tidal Heating

Corresponding Author: Mr Allard Veenstra

Version 0:

Reviewer comments:

Reviewer #1

(Remarks to the Author)
See attached.

[Editorial Note: This attachment is displayed at the end of the file]

(Remarks on code availability)

Reviewer #2

(Remarks to the Author)

Congratulations on completing and submitting your manuscript! I think the main finding of this paper, and the one I find most compelling, is that even in the absence of a global magma ocean, a feedback mechanism between melt fraction and tidal forcing can produce a longitudinal offset in heat flow distribution. This is an important outcome and is worth publishing. However, I do have major concerns with the framing and presentation of this model. I think the current version of this manuscript makes claims that I don't fully understand and that the results of the model may be over-interpreted.

First is a minor complaint, likely an overleaf/latex issue: there are multiple instances of "Figure ??" instead of a numbered figure reference, and there is a reference to a "Table ??" that doesn't seem to exist in this version of the manuscript.

A note about the data that's being compared to: the views that we have of Io are globally complete, but are discrete in time and represent only a few wavelength bands. The limitations of using only a few IR and near-IR bands to represent the entirety of Io's volcanic activity should be discussed in greater detail: i.e. you mention multiple observing campaigns from many observatories/spacecraft, but ultimately the dataset that you compare your model against uses only Juno data. This should be laid out clearly. Additionally, your model produces heat flow. You compare this heat flow to volcanic activity. Your weakly coupled cases, though, represent the distribution being spread-out or redirected by alternate heat transport (like convection). However, my understanding is that something like that would necessarily mean that the volcanic distribution is less correlated to surface heat flow. While you explain your choice to treat volcanic flux as a proxy for surface heat flow, I would like to see discussion of this assumption when you discuss the implications of the model.

The model: There are quite a few assumptions made in your model. I think they are, for the most part, reasonable. However, your argument for using an asthenospheric heating regime is circular; you assume an asthenospheric regime due to observations of volcanic flux, and then trying to match your model's distribution to the observations of volcanic flux. More discussion of the effects on your model would be appreciated.

A critical issue that I find with the paper is your claim that your model can reproduce the observed longitudinal offset. In line 263 you state that you can get a 15 degree eastward longitudinal shift (in the medium-c case), but the longitudinal shift in volcanic activity that we've observed is much higher than that. The studies you cite observe a 30-60 degree shift (which you mention in the introduction). However, you don't discuss the disagreement there; the potential explanation for the shift is still a relevant scientific discovery, but the results you're producing don't seem to match the observations, and your paper seems to claim they do. Furthermore, with higher-coupled models, you mention that the eastward shift cycles. This is extremely

interesting, but also raises some red flags for me-- first off, if the model is continuously changing in time, then how do you determine that it's converged? Do you have a different metric for determining convergence? How would we expect to measure this effect.

Now, unfortunately, I do have some major issues with the plots. In particular Figure 2 I feel does not support your claims very well. In your supplementary, the plot of NUMBER of volcanoes actually seems to match well with your modeled heat flow; this is interesting, but I would love to see discussion of the binning in these plots. However, unlike in the supplementary, the red line in Fig. 2 (observed volcanic flux) doesn't seem, to my eye, to match the distribution of any of your c-values-- and in fact, it doesn't seem to match any of the modeled data better or worse than it matches the uniform case. Since the interpretation of the shape of the model is not clear-cut, I think you need to include some form of fitting/error analysis to demonstrate that your model agrees with the data.

I'll reiterate that your fundamental finding-- that you can explain a longitudinal offset using a feedback mechanism between melt fraction and tidal forcing-- is scientifically interesting and worth publishing in this journal. However, I think at the minimum, the following major issues need to be addressed before I can recommend publication:

- 1) The model's fit to the data is, to my eye, tenuous, and I think a statistical comparison needs to be done.
- 2) Discussion of model convergence (how it's determined) and more robust/straightforward discussion of the model assumptions
- 3) Expanded discussion of the high-c offset cycling

Good luck with your edits! I hope the editing process goes smoothly, and I look forward to reading the revised manuscript.

(Remarks on code availability)

Version 1:

Reviewer comments:

Reviewer #1

(Remarks to the Author)

The authors have made numerous revisions to the MS and in my view it is now essentially ready for publication. I make a few additional suggestions below, but I don't need to see it again. This is a nice piece of work.

(Line numbers refer to the marked-up version)

l.234-235 I still think a reference to the Khurana et al. lower bound of 20% melt fraction is appropriate for the main text (though as the authors note its validity is disputed).

l.458 without a subsequent shift

l.517 Is the second significant figure really justified? And the number of iterations is not going to make sense to most people, so it probably doesn't want to be in the main text - more important is the conversion to an actual timescale (l. 530 below).

l.668 "due to spacecraft insensitivity" is a bit clumsy and also ignores the potential for telescopic observations. Does this phrase need to be here?

l.670 "a higher proportion of" rather than "an emphasis on"

l.713 This is a convoluted sentence. I would say "Finally, our model also presents a tentative explanation for any observed eastward shift that might be detected, namely that the conditions in Io are such that no stable tidal dissipation pattern arises".

l.718 one would expect

l.747 small polar dichotomy might be amplified

l.1021 Here C is a constant

l.1172 is found becomes lower

l.1201 "the former melt fraction" - does this mean the melt fraction from the previous timestep (i.e. a kind of Lax averaging)?

(Remarks on code availability)

Reviewer #2

(Remarks to the Author)

Wow, this paper has been hugely improved! It is really looking in excellent shape. I feel much better about the comparisons being made, the justification of the model parameters, and the conclusions being drawn. I think with very minor revisions it will be ready to publish.

I have one remaining gripe, which is on the comparison between the model results and the data. I don't think the Spearman rank correlation is an appropriate metric for a few reasons. (For clarity, I'm going to switch to calling it SRC). Since the SRC tracks monotonic agreement, of course the uniform distribution would have low SRC-- it neither increases nor decreases with longitude, and therefore is almost never going to be in monotonic agreement with the simulated data. However, in Supplementary Table 1, it looks like you're getting non-zero SRC values when doing 2D comparisons against the uniform case, which I'm a bit confused by. I'm also a bit concerned that while, to my eye, the model you've produced really seems in decent agreement with the number of hotspots, the SRC is lower than I'd expect in a well-fit model. Also, the bin sizes significantly change the SRC-- while yes, the strong-coupled case is still the highest regardless of bin size, a jump in .5 (for 2D feature) on a -1 to 1 scale based on a bin-size change is huge. Anyway, that's all to say-- I think SRC is inappropriate for comparing to 2-dimensional datasets.

I'd still love to see some statistical comparison between the model and the data. However, I'd say it's now of a much lower priority since the paper no longer claims to match the data well (only producing the offset).

With that complaint aside, I think the paper looks quite good. Thank you for taking our comments into consideration! Congratulations, and excellent work!

(Remarks on code availability)

Introduction

We want to thank both reviewers for their detailed and thoughtful comments. We have used their comments and suggestions to improve our manuscript.

Following, we answer each comment individually for both reviewers. We have taken the liberty of splitting some of the comments to keep our answers focused, but have not changed the order or the wording of the original comments. The original review text is in red, and our response is in black.

Reviewer #1

1a. **Figure 2.** The comparison between the longitudinally and latitudinally-averaged models and observations are not very convincing. None of the models show the observed 45 degree displacement in longitude, and none of them replicate the absence of a peak at ~180W. And the best (though it is not very good), “strong” model shows the worst fit to the latitudinal pattern. I have several comments on this.

Response:

The intention of the paper was not to produce a better fitting map for the observed heat flux or the latitudinally averaged observations, but to provide an explanation for the longitudinal shift, as recognized by Reviewer 2. We agree that part of our introduction, part of the discussion, and Figure 2 implied a more substantiated explanation of the observations than just the shift. To resolve this tension, we have changed the wording in both the introduction and discussion (see lines 93-97 and 520-525 respectively), and we have added a new section: 'comparison with observations', to better focus the comparison with the observations. The observed 45 degree displacement and the absence of a peak at ~180W are only observed in the distribution of volcanic flux; we added a discussion on that in the Discussion (see lines 569-590 and 599-628) and in our response to comments 1c and 1d.

1b. The authors should do a statistical comparison of the observations and data. The best way to do so is in the spherical harmonic domain, by comparing individual coefficients. For instance, there is an obvious (1,0) term in the data (the absence of a peak at 180W) that will not arise in any spherically symmetric models. The authors should calculate whether the feedback models actually provide a better fit to the coefficients than spherically-symmetric models do and if so, how much better.

Response:

We acknowledge the need for a more thorough statistical comparison, but have used binning to do so. We investigate the effect of uncertainty due to bin size for both observation and models, and the uncertainty due to the c-factor for the model. Furthermore, we also account for uncertainty in the interpretation of the observations by also showing the observed shift, assuming volcanic source numbers as a proxy for heat flow (as suggested by the reviewer in comment 2). The effect of all of these variations is shown visually in an extra sub-figure in Figure 2. We show that our model provides a better match for all cases, with the model with strong coupling performing best. The models show higher Spearman rank correlation (Supplementary Table 1, Supplementary Information) with the observed longitudinally averaged number density and somewhat higher for the longitudinally averaged heat flux

(shown below). We do not include the correlations with the heat flux in Supp. Table 1 because Figure 2 already demonstrates a poor fit. Finally, we expand the discussion of the continuously moving pattern that can, in principle, explain any longitudinal shift (see comment 2).

We disagreed with using spherical harmonics for the comparison because it has problems working with volcanic heat flux. The use of spherical harmonic decomposition works well with volcanic source density (e.g. Hamilton et al. (2013) & Steinke et al. (2020b)), but the same approach breaks down when looking at volcanic heat flux. The underlying mathematical assumption of point sources in the method described in Kirchoff et al. (2011) does not interface well with very large sources such as Loki Patera. To use harmonic decomposition on volcanic heat flux, additional assumptions are needed to smooth the data beforehand, as done in Pettine et al. (2024). These assumptions bring uncertainty because the smoothing should be linked to physical processes, such as what the magma plumbing system looks like and how big potential magma chambers are. We think this discussion would detract from the focus of the paper and be beyond its scope.

We assume the reviewer means a (1,1) term in the observed volcanic heat flux. We discuss this dichotomy in our response to comment 1d.

Bin size [deg]:	5	10	15	20	36
Uniform model, Longitudinal Flux	-0.199	-0.338	-0.559	-0.557	-0.636
Medium coupling, Longitudinal Flux	-0.070	-0.206	-0.535	-0.431	-0.172
Strong coupling, Longitudinal Flux	-0.007	-0.108	-0.385	-0.390	-0.172

Table 1reb: *The Spearman rank correlations for the longitudinally averaged heat flux.*

1c. **The observed 45 degree displacement is presumably the maximum this model could produce (because librations are symmetrical around the 45 degree line). So is there some mechanism that drives the system to this, presumably stable, point?**

Response:

The observed 45 degree displacement only occurs in the volcanic heat flux distribution on the leading hemisphere. Therefore, it is not something that our model should necessarily generate. It also cannot. The largest offset for a stable, converging system that we found is around 20 degrees.

We ran simulations starting with a degree 2, order 2 (2,2) melt fraction distribution with a 45 degree shift, to investigate whether it might be a stable point. We find that the initial shift and additional amplitude in the melt fraction quickly go away. We eventually end up in the same end state as if we had started with a uniform interior. This is an interesting result that is somewhat predicted by the toy model that we present in the article (Fig.3). The end-state of

the system following the feedback mechanism in the current model is solely determined by the value of the coupling coefficient c . We have also mentioned this in the article (see lines 456-459).

To further illustrate that Fig. 3 in the article predicts this, we modified it by assuming a certain value for c (Fig. 2reb below). This allows us to fully map the input to the output by representing the output as a vector with its orientation determined by the eastward shift and amplitude of the **resulting** melt fraction pattern. In this way, all possible iteration trajectories with that c value are shown. The plot below (Fig. 2reb) shows that no matter where you start in this state space, you will end up at the same point on the stable line.

Fig. 1reb. Demonstration of how an initial (2,2) melt fraction pattern, with a 45-degree displacement, is not stable. We used $c=0.011$ (strong strength), and the outcome is the same as the medium c -factor case in Fig.2 of the manuscript.

Fig. 2reb. Coupling the input state to the output state for a given value for c ($c=0.015$) shows that there is a single stable point. The size of the vectors is not to scale. The yellow line does not follow the vectors due to the averaging of new melt fraction patterns with old ones.

1d. Similarly, is there any way a (1,0) term arising in the models could be amplified? Kang & Flierl (2020) propose a way in which such a term can arise at Enceladus. The MS would be more convincing if the model provided a better match to the observations. In particular, the absence of any peak at $\sim 180^\circ$ is troubling and (despite the authors' attempts to wave it away on I.495ff) makes it harder to believe that this model really represents what is happening inside Io.

Response:

We assume the reviewer refers to the (1,1) pattern evident in the heat flux observations. As we now include a comparison with volcanic source distribution, there is a peak at 180° , indicating that the absence of a peak in the observed heat flux might be due to the existence of hard-to-detect sources. We now cite intrusive and stealth volcanism as processes that can make heat output difficult to detect (see lines 605-616). In the article, we mention that investigating where this dichotomy, or lack of a peak at $\sim 180^\circ$, comes from is something we would like to investigate using our model (see lines 627-629). As a potential prelude to this, and because we agree with the reviewer that the stability or amplification of a (1,0) or (1,1) dichotomy is very interesting, we ran some additional simulations.

We ran simulations that started with a (1,1) melt fraction pattern, with its maximum shifted to be at 270° west (see Fig. 3reb.). Iterating the feedback mechanism with these starting conditions, we again see that the additional modes die off, and the model eventually returns to a pattern that we always get for a given c -factor. We also looked at a (1,0) in the asthenosphere and found that it, too, is not self-sustaining.

Overall, this analysis shows that both types of dichotomies are not self-sustaining via the melt tidal heating mechanism investigated here. Other non-modelled processes or structural differences within Io (e.g., mantle plume, lithospheric thickness variations) might explain this dichotomy.

Fig. 3reb. Demonstration of how an initial (1,1) melt fraction pattern, with a 90 degree displacement, is not stable. We again used $c=0.011$ (strong strength), and the outcome would have led to the same c -factor case in Fig.2 of the manuscript, only we stopped the iteration prematurely to save some time.

2. I think the authors should spend more space discussing the possibility that what we are seeing is not a stable pattern but a time-variable one. The calculated evolution timescale (~ 1 Myr) is very similar to the estimated surface age of Io (~ 1 Myr) and the estimated values of c favour a time-variable case (1.687). How would we tell? One way would be to look at whether the distribution of volcanoes and the estimated heat flux differ – the latter is a snap-shot at the present day while the former have been accumulating over 1 Myr. I'm not an expert, but if you look at Fig 1 of Steinke et al. (2020) you see a peak in equatorial volcano density at 120 degrees W, which is not seen in the heat flux distribution (nor in Fig A3). Assuming the Steinke data are OK, that suggests to me that we are seeing a continuously evolving situation and not a static one.

Response:

We agree with the suggestion that the section about the time-variable case should be expanded. The discussion of the effects of a non-static Io has expanded under a new section called: “A continuously moving dissipation pattern”, and we have incorporated it more into Fig. 2 and the results section. We would also like to thank the reviewer for the suggestion of how the existence of a continuously evolving situation could be tested. We have added a short discussion, based on the results of Hamilton et al. (2013) and Steinke et al. (2020b), and what could be done next, to the article (see lines 645-658).

3a. How good are the data? It would be good to explain how the heat flux map shown in Fig 2 is derived, especially as it is quite deceptive – the real heat flux distribution is much patchier, as can be seen in the equivalent heat flux map shown in Pettine et al. (2020), Fig 2.

Response:

Yes, we agree that the quality of the data requires more discussion. We have changed Fig. 2 to show the hotspot marker size as a function of energy emitted. Furthermore, as mentioned in our response to 1b and 3b, we made the comparison more robust by the use of volcanic feature density as well as volcanic heat flux when linking observations to model outcomes.

3b. The extreme patchiness is also a warning that maybe using the present-day snapshot represented by heat flow is not a good idea, and that using volcano density instead (or in addition) is something the authors should seriously consider. I do note that the longitudinal distribution shown in the Pettine et al. plot is similar to that shown in this MS (for instance, both show an absence of heat at 180W).

Response:

We agree and have added the comparison with volcano density in Figure 2, the results section, and the discussion and conclusion.

4a. The parameter c. I found the use of this parameter quite obscure and I think the authors should add some further derivation. The way I thought about it was that you can balance the melt production rate (heat production/latent heat) against the melt migration rate (which depends on the permeability, buoyancy etc. – see Moore 2001). If you do that then you end up with an equilibrium melt fraction ϕ that depends on the local heat production h (in Wm^{-3}) as

$$\phi^{(n+1)} = Ch$$

where C is a constant and the $n+1$ comes from the fact that permeability goes as ϕ^n . If you differentiate both sides you end up with

$$d\phi = \phi / (n+1) dh/h$$

which is eq. 1 if you swap Q (Wm^{-2}) for h (Wm^{-3}). Since n is 2 or 3, the constant of proportionality c between $d\phi$ and dQ is roughly 0.015, consistent with the MS's numbers.

Response:

We are very grateful to the reviewer for pointing us to this alternative derivation and have incorporated it into the article (see lines 880-894). However, when we did the above derivation ourselves, we ended up without the +1, so: $d\phi = \phi / (n) dh/h$. Looking through the literature, like the appendix in Spencer et al. (2020a), we also did not find the $(n+1)$. We added more explanation where the c -factor comes from (see lines 861-878).

4b. It took me a while to figure out that the MS uses Wm^{-2} as its units for Q (it never says this). Also, the expression for the iteration timescale on I.404 is just wrong. And c is quoted without units, which is also incorrect (it has units of m^2W^{-1}).

Response:

We added the units to both Q (see line 798) and c (see lines 208 and 813). We also fixed the expression for the iteration timescale, which was indeed wrongly stated, but had correct estimates for the time.

5. Predictions. One of the most useful things this MS could do is to make predictions which are potentially testable with future ground-based observations (or a future mission). For instance, in principle a non-spherically-symmetric structure would give rise to unexpected tidal responses, as has been predicted for the Moon (Zhong & Qin 2012). Similarly, if one could map out the melt distribution (e.g. via induction) one would see a structure that is not symmetric about the meridian. Maybe there's something about spatial variations in melt temperature? Anyway, the authors should give some thought to this.

Response:

We thank the reviewer for their suggestions. One testable hypothesis we have now included in the paper is related to the time-varying tidal dissipation pattern, following the earlier suggestion of the reviewer (see lines 645-658).

We have also looked at the effect of the lateral rheology variations on the tidal response, as suggested. The pattern produced by a strong coupling (Fig. 2 in the article) has a difference of $\sim 1\%$ between k_{20} and k_{22} , while for a uniform model, these two love numbers are identical. This difference is, for instance, above the sensitivity of JUICE, which will orbit Ganymede but is on the edge of the predicted sensitivity of Europa Clipper (or IVO). Higher order tidal responses are on the order of 0.2%. We have added this suggestion/prediction to the discussion (see lines 630-641) and want to thank the reviewer for their suggestion.

Another option, as the reviewer suggests, is to verify our prediction of a laterally varying melt distribution using future induction measurements. However, we feel we are currently not yet at the stage of predicting anything other than that the distribution would not be symmetric about the meridian. With future model improvements, especially to the thermal transport, this is worth considering.

Minor comments:

I.137 There are other authors who have codes with similar capabilities. Zhong and Qin (2012) certainly do, and there are probably others (e.g. Harriet Lau's group).

Response:

We agree, we have added references to two papers of Qin and one of Harriet Lau's group (see line 157). However, we note here that those codes used perturbation theory, which requires the lateral variations to be small. The lateral viscosity variations arising from the feedback mechanism in our study are outside this regime. Currently, only LOV3D or a FEM code can tackle our problem.

I.179ff This description is pretty obscure – refer the reader to the Methods

Response:

Agree, we've added the reference to the Methods.

I.196 Values for c are not given in section 2 – they are in the Methods.

Response:

Agree, we changed the reference.

I.302 The 45 degree shift is nicely explained by Murray and Dermott in their chapter on tides.

Response:

We added the reference.

Fig 3. In the caption, the axis label descriptions are switched.

Response:

This has been fixed.

Fig 3. I did not find this figure easy to interpret. It might be easier if the y-axis showed the total shift, not the initial shift (because the total shift is what is actually observed).

Response:

To improve the clarity of the figure, we (1) extended the description in the caption, (2) expanded the description in the text (see lines 428-440), (3) modified the axis labels to better reflect what is happening, and (4) coloured and named the trajectories that are plotted in the figure.

The way the figure is currently constructed allows it to map input to output, within the confines of the toy model. A (2,2) melt fraction pattern can be fully determined by two parameters: its amplitude and its eastward shift. The tidal dissipation resulting from those lateral variations is also characterized by an eastward shift. So by mapping the starting amplitude and eastward shift to the resulting eastward shift, we see how the system responds to different inputs. Changing the y-axis to total shift would lose the information about the initial pattern.

Fig 3. Also the “increasing c ” arrow makes it look as if c is increasing along an individual black line, which is not what the authors are intending.

Response:

We have removed the arrow and instead put the measure of strength in the line. The same goes for the stable line. We have also added a small arrow on top of the line to indicate the direction with iteration.

I.409 Millions of years is roughly the surface age of Io. So we could be seeing the consequences of a continuous eastwards drift.

Response:

We have added a mention of this to the article (see lines 498-502).

I.476 This seems like a bold claim, given the rheological simplifications and neglect of deep tidal heating. It would be more realistic to say “is unlikely to be affected by our model choices”.

Response:

We agree, the wording has been changed to what was suggested.

I.515 I don't understand why a N-S dichotomy would produce a leading-trailing hemisphere dichotomy. It is a better idea to quote the observations of (e.g.) De Kleer et al. 2016 Fig 3 who show that there is a different pattern of eruption characteristics on the leading vs. the trailing hemisphere. That strongly suggests some kind of degree-1 variation in lithospheric properties.

Response:

Agreed, this is a non-sequitur - the N-S dichotomy reference has been removed, and we focus on the dichotomy between leading and trailing hemispheres. We added the reference to de Kleer et al. (2016).

L.535 I agree with this suggestion – but it has already been developed, by Kang & Flierl (2020).

Response:

Thank you for pointing us to this research; we missed this. We have rewritten the text to incorporate it (see lines 671-676).

I.586 The assumption of incompressibility is fine, but the Maxwell assumption is questionable. A better model would be to use an Andrade rheology. The authors should at least briefly discuss whether they think the results will change if they did so.

Response:

We agree with the reviewer that, given the measurement of k_2 by Park et al. (2024) and previous publications, Andrade rheology is a better description of the rheology. We have added a discussion on the use of Andrade rheology (see lines 724-736). We do not expect the conclusion to change significantly with the use of Andrade rheology. For our purposes, with only one relevant timescale, the only effect of using Andrade rheology would be the use of more realistic values for viscosity and shear modulus.

I.631 Give units for Q.

Response:

Added the units

I.658 “average” means “spatially-averaged”, right?

Response:

Yes you are right, we changed to wording to make this clear.

I.661 conformable to our assumption

Response:

Changed the wording

I.674 The 10% melt fraction assumption is low compared with that of Khurana et al. (2011) who found a lower bound of 20%. This is worth mentioning.

Response:

We have added a mention to Khurana et al. (2011) and a short discussion in the text (see lines 779-790).

I.765ff This section made me uncomfortable. The methods outlined here effectively prevent the development of any kind of thermal runaway (which is presumably responsible for lo heating up in the first place). The argument presented in the MS is that melting is so effective at heat removal that the total heat production never changes. But this is not really consistent with the model's assumption that local heat production is directly dependent on changes in melt fraction (equation 1). Since the authors find that overall the same qualitative behavior is obtained (I.786) I don't think this is a show-stopper. But I would like to see an extra plot like Fig 2 but without the normalization included, because I suspect that is a more realistic picture of what is happening.

Response:

Initially, we normalized the heat flux because the c -factor depends on the average melt fraction and average tidal heating (see the relation in comment 4a). However, this has the inconvenience that, as pointed out by the reviewer, thermal runaway is prevented. Because c in our model only approximates more complex melt transport, we opted not to normalize the heat flux to allow for thermal runaway. We note that this model choice does not affect our general conclusions. We have updated Fig. 2 in the article to be not normalized, and as expected, it is very similar to the old, normalized version. The old, normalized version has been moved to the appendix (see Supplementary Fig. 7).

This model change does mean that the average melt fraction is no longer connected to average heat flux, and that the average dissipation increases. We have mentioned this in the article (see lines 949-955).

Reviewer #2

1. A note about the data that's being compared to: the views that we have of Io are globally complete, but are discrete in time and represent only a few wavelength bands. The limitations of using only a few IR and near-IR bands to represent the entirety of Io's volcanic activity should be discussed in greater detail: i.e. you mention multiple observing campaigns from many observatories/spacecraft, but ultimately the dataset that you compare your model against uses only Juno data. This should be laid out clearly.

Response:

The hot spot dataset that was published in the cited PSJ paper (Davies et al., 2024b), which we use to compare to the heat flow model, is not limited to *Juno* data. The hot spot database includes data from the *Galileo* Near Infrared Mapping Spectrometer (NIMS) and Photo-Polarimeter Radiometer (PPR) instruments collected between 1996 and 2002; from the *New Horizons* spacecraft (2007); from ground-based telescopes (IRTF, Keck, Gemini, LBT); and from *Juno*/JIRAM. The PPR, and some IRTF data, are at wavelengths that extend into the thermal infrared. The hot spot thermal emission dataset we compare our model results to is derived from a compilation of measurements collected over more than thirty years. *Juno*/JIRAM are the only data we have for the polar hot spot detections, but these still span more than seven years.

To reflect the above, the text has been modified to read (see lines 188-194): “The hot spot thermal emission database was compiled using both spacecraft and ground-based telescope data collected over more than three decades, and *Juno* data collected over seven years. The database includes hot spots that are not easily detectable by JIRAM (Davies et al., 2024).”

2. Additionally, your model produces heat flow. You compare this heat flow to volcanic activity. Your weakly coupled cases, though, represent the distribution being spread-out or redirected by alternate heat transport (like convection). However, my understanding is that something like that would necessarily mean that the volcanic distribution is less correlated to surface heat flow. While you explain your choice to treat volcanic flux as a proxy for surface heat flow, I would like to see discussion of this assumption when you discuss the implications of the model.

Response:

It is true that for a weakly coupled case our model implies that the volcanic heat flux/distribution is less correlated to surface heat flow. However, we would like to stress that we use only the long-wavelength part of the observed volcanic activity by considering binsizes between 5 and 36 degrees. At these large wavelengths, we assume that even a weak coupling would still lead to a correlation between tidal heating and volcanic activity.

We have added more discussion on using volcanic flux as a proxy for the tidal dissipation inside Io (see lines 317-327 and 569-590). Our original manuscript only used volcanic flux. To make the comparison less dependent on the assumptions for the proxy, we also include a comparison to the distribution of volcanic features; see also our response to comment 1b. by R#1.

3. The model: There are quite a few assumptions made in your model. I think they are, for the most part, reasonable. However, your argument for using an asthenospheric heating regime is circular; you assume an asthenospheric regime due to observations of volcanic flux, and then trying to match your model's distribution to the observations of volcanic flux. More discussion of the effects on your model would be appreciated.

Response:

We agree that there should be more discussion on the effect of the starting interior model and have added it to the article. The role of deep mantle heating is discussed in the discussion and methods (see lines 532-541 and 747-754 respectively), and the role of asthenosphere properties different than those assumed is expanded upon (see lines 737-745). However, both things do not change our conclusion of an eastward shift.

Coming back to the issue of circular reasoning raised by the reviewer. To start the iterations, an initial interior structure needs to be specified. We decided to consider a model dominated by asthenosphere tidal heating for two reasons: (1) this is both consistent with enhanced heat flux at low and mid-latitudes and (2) expected from thermal models (Spencer et al. (2020b)). Moreover, the focus on the asthenosphere also follows from the fact that the feedback mechanism under investigation is likely stronger in this layer (partial melt has a larger effect on viscosity variations than temperature).

4. A critical issue that I find with the paper is your claim that your model can reproduce the observed longitudinal offset. In line 263 you state that you can get a 15 degree eastward longitudinal shift (in the medium-c case), but the longitudinal shift in volcanic activity that we've observed is much higher than that. The studies you cite observe a 30-60 degree shift (which you mention in the introduction). However, you don't discuss the disagreement there; the potential explanation for the shift is still a relevant scientific discovery, but the results you're producing don't seem to match the observations, and your paper seems to claim they do.

Response:

We agree that parts of the paper suggested a fit to the observations that the results do not warrant, see also our response to comment 1a for R#1. We have addressed this in multiple ways. Firstly, we have changed the wording in the introduction and discussion to reflect a more modest interpretation (see lines 93-97 and 521-525 respectively). Secondly, we include a more rigorous comparison between the model and the observations: we include the volcanic feature distribution more prominently, we vary the bin size, and we compare to different c-factors (see lines 330-348 and Supplementary Table 1). Thirdly, we discuss the mismatch between our model and the observations and emphasize that an alternative explanation is that we see a non-stable heat flux pattern (see lines 339-348 and 645-658 respectively), and see also comment 5c below.

5a. Furthermore, with higher-coupled models, you mention that the eastward shift cycles. This is extremely interesting, but also raises some red flags for me-- first off, if the model is continuously changing in time, then how do you determine that it's converged?

Response:

What we see is that the model displays periodic behavior. The peak dissipation continuously moves eastward, but the patterns repeat every x iterations (the exact number depends on the coupling strength, for the toy model it is 110 (see line 478)). For example, the pattern shown in Supplementary Fig. 4 at iteration number 25 is repeated in iteration 134 (we have changed this figure as a reaction to this comment). We investigated this further by looking at the average dissipation and the value of relevant energy modes. Both show a cyclical behavior, consistent with switching between patterns.

Supplementary Figure 6 also shows this cyclical behaviour in the toy model. It shows how the eastward shift vs the size of the variations, which in the toy model actually determines the entire system, evolves with iteration. The line returns to itself and overlaps, showing that the system is cyclical.

5b. **Do you have a different metric for determining convergence?**

Response:

In the static case, we linked convergence to changes in dissipation. We check whether the average energy no longer changes as the basis for convergence.

5c. **How would we expect to measure this effect.**

Response:

One way to perhaps test the hypothesis of a non-static tidal dissipation is to see if there is a significant difference in distribution between active hotspots and old hotspots. Results in a previous paper (Steinke et al. (2020b), compare Fig. 1 with Fig. A1), suggest that something like this might be happening, as suggested by reviewer 1. We have added a section on this idea in the article (see lines 645-658).

6. **Now, unfortunately, I do have some major issues with the plots. In particular Figure 2 I feel does not support your claims very well. In your supplementary, the plot of NUMBER of volcanoes actually seems to match well with your modeled heat flow; this is interesting, but I would love to see discussion of the binning in these plots. However, unlike in the supplementary, the red line in Fig. 2 (observed volcanic flux) doesn't seem, to my eye, to match the distribution of any of your c-values-- and in fact, it doesn't seem to match any of the modeled data better or worse than it matches the uniform case. Since the interpretation of the shape of the model is not clear-cut, I think you need to include some form of fitting/error analysis to demonstrate that your model agrees with the data.**

Response:

We thank the reviewer for their suggestion. We have now updated Fig. 2 to include the volcanic feature numbers as well as the volcanic flux. Furthermore, we've added a plot showing the offset for the volcanic feature density, the volcanic heat flux, and the three models for different bin sizes. This immediately shows whether our model matches the observations in terms of longitudinal shift. We varied the bin size to investigate its effect on the results and show that our finding of a better explanation for the offset is robust. The results, as represented in the new Fig. 2, show that the best match with the observations for all bin sizes is the strongly coupled model. Finally, we also emphasize the tentative explanation for any observed shift via a non-static pattern.

We compute the Spearman rank correlation and find that, compared to the uniform case, all correlations are higher. This finding is irrespective of bin size. The correlations can be found in Supplementary Table 1 and in the table in our response to comment 1b of R#1. At the largest shown binsize (36 degrees), the correlation is relatively high (~0.5), a sign that we can reasonably match the observed eastward shift in volcanic features. As we mention in the header text for Supp. Table 1, the correlations change if you alter the bin spacing. In Fig. 2 and for the correlation in Supp. Table 1 we make sure that the centre of the last bin is on the prime meridian (360W). For a more traditional bin spacing, where the centre of the first bin is located at 0 + half the binsize, the previously given correlation becomes ~0.7. However, the relative correlations between our model and the uniform model do not change.

We discuss the above in the new section: "Comparison with observations".

7a. **Last points.** I'll reiterate that your fundamental finding-- that you can explain a longitudinal offset using a feedback mechanism between melt fraction and tidal forcing-- is scientifically interesting and worth publishing in this journal. However, I think at the minimum, the following major issues need to be addressed before I can recommend publication:
1) The model's fit to the data is, to my eye, tenuous, and I think a statistical comparison needs to be done.

Response:

We believe we answered this point in our response to comments 4 and 6.

7b. **2) Discussion of model convergence (how it's determined) and more robust/straightforward discussion of the model assumptions**

Response:

We feel model convergence is adequately addressed by our response to comments 5a and 5b. The concerns for the model assumptions we have also addressed in our response to comment 3 and in general: (1) we have updated the method section to clarify our assumptions. (2) We have expanded our discussion on the limitations and relevance of our assumptions. For instance, our choice of assuming asthenospheric heating (see lines 737-745), or the limitations of the thermal model employed (see 532-545)

7c. **3) Expanded discussion of the high-c offset cycling**

Response:

We addressed this in our response to comments 5a and 5c and comment 2 of R#1.

REVIEWERS' COMMENTS

We thank the two reviewers for their thoughtful comments. We are happy to see both reviewers appreciate the changes we made to the manuscript.

Reviewer #1 (Remarks to the Author):

The authors have made numerous revisions to the MS and in my view it is now essentially ready for publication. I make a few additional suggestions below, but I don't need to see it again. This is a nice piece of work.

(Line numbers refer to the marked-up version)

I.234-235 I still think a reference to the Khurana et al. lower bound of 20% melt fraction is appropriate for the main text (though as the authors note its validity is disputed).

Response:

We add the reference and have merged the discussion on average melt fraction that was in the Method section with the one in the Results section, (lines 226-238)

I.458 without a subsequent shift

Response:

Corrected

I.517 Is the second significant figure really justified? And the number of iterations is not going to make sense to most people, so it probably doesn't want to be in the main text - more important is the conversion to an actual timescale (I. 530 below).

Response:

We agree, we have removed the explicit mention of the number.

I.668 "due to spacecraft insensitivity" is a bit clumsy and also ignores the potential for telescopic observations. Does this phrase need to be here?

Response:

Agreed, we have removed the phrase

I.670 "a higher proportion of" rather than "an emphasis on"

Response:

Agreed, we have changed the text

I.713 This is a convoluted sentence. I would say "Finally, our model also presents a tentative explanation for any observed eastward shift that might be detected, namely that the conditions in Io are such that no stable tidal dissipation pattern arises".

Response:

We agree that the sentence was unclear and replaced it with the suggested sentence without "that might be detected": "Finally, our model also presents a tentative explanation for

any observed eastward shift, namely that the conditions in Io are such that no stable tidal dissipation pattern arises.”

I.718 one would expect

Response:

Corrected

I.747 small polar dichotomy might be amplified

Response:

Corrected

I.1021 Here C is a constant

Response:

Corrected

I.1172 is found becomes lower

Response:

No, when we normalize, the value for c to obtain a certain shift increases. This can be seen by comparing Supplementary Fig. 7 and Fig. 2; the plotted maps look similar but to produce Supplementary Fig. 7, we used $c=0.015$ for the strong model, while for Fig. 2, we used $c=0.0105$.

We have changed the text to: “the value for c at which a certain eastward shift is found **generally increases**” (see line 946).

I.1201 “the former melt fraction” - does this mean the melt fraction from the previous timestep (i.e. a kind of Lax averaging)?

Response:

Yes, we have changed the text to say “the melt fraction distribution from the previous iteration” instead of “former” to make this clearer (see line 962-965).

Reviewer #2 (Remarks to the Author):

Wow, this paper has been hugely improved! It is really looking in excellent shape. I feel much better about the comparisons being made, the justification of the model parameters, and the conclusions being drawn. I think with very minor revisions it will be ready to publish.

Thank you for the kind words.

I have one remaining gripe, which is on the comparison between the model results and the data. I don't think the Spearman rank correlation is an appropriate metric for a few reasons. (For clarity, I'm going to switch to calling it SRC). Since the SRC tracks monotonic agreement, of course the uniform distribution would have low SRC-- it neither increases nor decreases with longitude, and therefore is almost

never going to be in monotonic agreement with the simulated data. However, in Supplementary Table 1, it looks like you're getting non-zero SRC values when doing 2D comparisons against the uniform case, which I'm a bit confused by.

Response:

We believe there is a misunderstanding about what we refer to as a uniform model. We use the term to refer to an interior model of Io with uniform properties at a given depth, as opposed to laterally varying properties. However, the tidal dissipation pattern produced by such a uniform interior model is distinctly not uniform (see Supplementary Fig. 1), which is why the SRC has non-zero values. To avoid this confusion We have changed “uniform model” to “spherically symmetric model” in the text and in Supplementary Table 1.

I'm also a bit concerned that while, to my eye, the model you've produced really seems in decent agreement with the number of hotspots, the SRC is lower than I'd expect in a well-fit model. Also, the bin sizes significantly change the SRC-- while yes, the strong-coupled case is still the highest regardless of bin size, a jump in .5 (for 2D feature) on a -1 to 1 scale based on a bin-size change is huge. Anyway, that's all to say-- I think SRC is inappropriate for comparing to 2-dimensional datasets.

Response:

The jump in correlation is due to a mismatch in spatial frequency content between the data and the model. The observations are a collection of point sources which can have high variability on small scales. The model data, on the other hand, only simulates long-wavelength features. For the strong coupling case, there is only significant dissipation up to degree 12 (equivalent to a bin size of 15 degrees), but most of the energy is found at degrees 2, 4 and 6. At bin sizes smaller than 15 degrees, the computed correlation is therefore not representative. We have removed the smaller bin sizes from Supplementary Table 1.

I'd still love to see some statistical comparison between the model and the data. However, I'd say it's now of a much lower priority since the paper no longer claims to match the data well (only producing the offset).

Response:

We think that the visual representation of the shift in model and data in Figure 2 is suitable to show the effect of the main uncertainties on the offset, and the table gives a quantitative comparison. Statistical comparisons of the 2D pattern are affected by several issues that are outside the realm of the model and we therefore decided to not include them.

With that complaint aside, I think the paper looks quite good. Thank you for taking our comments into consideration! Congratulations, and excellent work!

This interesting MS makes the case that the observed eastwards displacement of volcanoes on the surface of Io could be the result of a feedback between melting and tidal dissipation in the solid interior. Previously the displacement was attributed to magma ocean processes, but recent results suggest that a magma ocean is not in fact present. So this MS is intriguing in that it provides a self-consistent model that can potentially explain all the current observations.

I think the MS is in principle publishable, but requires some significant revisions first. My main concern is with how well (or not) the model fits the observations.

Figure 2. The comparison between the longitudinally and latitudinally-averaged models and observations are not very convincing. None of the models show the observed 45° displacement in longitude, and none of them replicate the *absence* of a peak at $\sim 180^\circ$ W. And the best (though it is not very good), “strong” model shows the worst fit to the latitudinal pattern. I have several comments on this.

1. The authors should do a statistical comparison of the observations and data. The best way to do so is in the spherical harmonic domain, by comparing individual coefficients. For instance, there is an obvious (1,0) term in the data (the absence of a peak at 180° W) that will not arise in any spherically symmetric models. The authors should calculate whether the feedback models actually provide a better fit to the coefficients than spherically-symmetric models do and if so, how much better.

The observed 45° displacement is presumably the maximum this model could produce (because librations are symmetrical around the 45° line). So is there some mechanism that drives the system to this, presumably stable, point? Similarly, is there any way a (1,0) term arising in the models could be amplified? Kang & Flierl (2020) propose a way in which such a term can arise at Enceladus. The MS would be more convincing if the model provided a better match to the observations. In particular, the absence of any peak at $\sim 180^\circ$ W is troubling and (despite the authors’ attempts to wave it away on 1.495ff) makes it harder to believe that this model really represents what is happening inside Io.

2. I think the authors should spend more space discussing the possibility that what we are seeing is not a stable pattern but a time-variable one. The calculated evolution timescale (~ 1 Myr) is very similar to the estimated surface age of Io (~ 1 Myr) and the estimated values of c favour a time-variable case (1.687). How would we tell? One way would be to look at whether the distribution of volcanoes and the estimated heat flux differ – the latter is a snap-shot at the present day while the former have been accumulating over 1 Myr. I’m not an expert, but if you look at Fig 1 of Steinke et al. (2020) you see a peak in equatorial volcano density at 120° W, which is not seen in the heat flux distribution (nor in Fig A3). Assuming the Steinke data are OK, that suggests to me that we are seeing a continuously evolving situation and not a static one.

3. How good are the data? It would be good to explain how the heat flux map shown in Fig 2 is derived, especially as it is quite deceptive – the real heat flux distribution is much patchier, as can be seen in the equivalent heat flux map shown in Pettine et al. (2020), Fig 2. The extreme

patchiness is also a warning that maybe using the present-day snapshot represented by heat flow is not a good idea, and that using volcano density instead (or in addition) is something the authors should seriously consider. I do note that the longitudinal distribution shown in the Pettine et al. plot is similar to that shown in this MS (for instance, both show an absence of heat at 180W).

The parameter c . I found the use of this parameter quite obscure and I think the authors should add some further derivation. The way I thought about it was that you can balance the melt production rate (heat production/latent heat) against the melt migration rate (which depends on the permeability, buoyancy etc. – see Moore 2001). If you do that then you end up with an equilibrium melt fraction ϕ that depends on the local heat production h (in Wm^{-3}) as

$$\phi^{n+1} = Ch$$

where C is a constant and the $n+1$ comes from the fact that permeability goes as ϕ^n . If you differentiate both sides you end up with

$$d\phi = \frac{\phi}{n+1} \frac{dh}{h}$$

which is eq. 1 if you swap Q (Wm^{-2}) for h (Wm^{-3}). Since n is 2 or 3, the constant of proportionality c between $d\phi$ and dQ is roughly 0.015, consistent with the MS's numbers.

It took me a while to figure out that the MS uses Wm^{-2} as its units for Q (it never says this). Also the expression for the iteration timescale on 1.404 is just wrong. And c is quoted without units, which is also incorrect (it has units of m^2W^{-1}).

Predictions. One of the most useful things this MS could do is to make predictions which are potentially testable with future ground-based observations (or a future mission). For instance, in principle a non-spherically-symmetric structure would give rise to unexpected tidal responses, as has been predicted for the Moon (Zhong & Qin 2012). Similarly, if one could map out the melt distribution (e.g. via induction) one would see a structure that is not symmetric about the meridian. Maybe there's something about spatial variations in melt temperature? Anyway, the authors should give some thought to this.

Other comments are more minor.

For some reason all Figure and Table references seem to have been lost (they are marked ??).

It would be helpful to have the Table in the main text if possible. Also, the Supplementary Information is useful, but if it is going to include figures they should be referred to in the main text (otherwise people won't read it).

1.137 There are other authors who have codes with similar capabilities. Zhong and Qin (2012) certainly do, and there are probably others (e.g. Harriet Lau's group).

1.179ff This description is pretty obscure – refer the reader to the Methods

1.196 Values for c are not given in section 2 – they are in the Methods.

1.302 The 45° shift is nicely explained by Murray and Dermott in their chapter on tides.

Fig 3. In the caption, the axis label descriptions are switched. I did not find this figure easy to interpret. It might be easier if the y-axis showed the total shift, not the initial shift (because the total shift is what is actually observed). Also the “increasing c ” arrow makes it look as if c is increasing along an individual black line, which is not what the authors are intending.

1.409 Millions of years is roughly the surface age of Io. So we could be seeing the consequences of a continuous eastwards drift.

1.476 This seems like a bold claim, given the rheological simplifications and neglect of deep tidal heating. It would be more realistic to say “is unlikely to be affected by our model choices”.

1.515 I don’t understand why a N-S dichotomy would produce a leading-trailing hemisphere dichotomy. It is a better idea to quote the observations of (e.g.) De Kleer et al. 2016 Fig 3 who show that there is a different pattern of eruption characteristics on the leading vs. the trailing hemisphere. That strongly suggests some kind of degree-1 variation in lithospheric properties.

L.535 I agree with this suggestion – but it has already been developed, by Kang & Flierl (2020).

1.586 The assumption of incompressibility is fine, but the Maxwell assumption is questionable. A better model would be to use an Andrade rheology. The authors should at least briefly discuss whether they think the results will change if they did so.

1.631 Give units for Q .

1.658 “average” means “spatially-averaged”, right?

1.661 conformable to our assumption

1.674 The 10% melt fraction assumption is low compared with that of Khurana et al. (2011) who found a lower bound of 20%. This is worth mentioning.

1.765ff This section made me uncomfortable. The methods outlined here effectively prevent the development of any kind of thermal runaway (which is presumably responsible for Io heating up in the first place). The argument presented in the MS is that melting is so effective at heat removal that the total heat production never changes. But this is not really consistent with the model’s assumption that local heat production is directly dependent on changes in melt fraction (equation 1). Since the authors find that overall the same qualitative behavior is obtained (1.786) I don’t think this is a show-stopper. But I would like to see an extra plot like Fig 2 but without the normalization included, because I suspect that is a more realistic picture of what is happening.